# Kernel Mean Estimation via Spectral Filtering

**Krikamol Muandet**
MPI-IS, Tübingen
krikamol@tue.mpg.de

**Bharath Sriperumbudur**
Dept. of Statistics, PSU
bks18@psu.edu

**Bernhard Schölkopf**
MPI-IS, Tübingen
bs@tue.mpg.de

## Abstract

The problem of estimating the kernel mean in a reproducing kernel Hilbert space (RKHS) is central to kernel methods in that it is used by classical approaches (e.g., when centering a kernel PCA matrix), and it also forms the core inference step of modern kernel methods (e.g., kernel-based non-parametric tests) that rely on embedding probability distributions in RKHSs. Previous work [1] has shown that shrinkage can help in constructing "better" estimators of the kernel mean than the empirical estimator. The present paper studies the consistency and admissibility of the estimators in [1], and proposes a wider class of shrinkage estimators that improve upon the empirical estimator by considering appropriate basis functions. Using the kernel PCA basis, we show that some of these estimators can be constructed using spectral filtering algorithms which are shown to be consistent under some technical assumptions. Our theoretical analysis also reveals a fundamental connection to the kernel-based supervised learning framework. The proposed estimators are simple to implement and perform well in practice.

## 1 Introduction

The kernel mean or the mean element, which corresponds to the mean of the kernel function in a reproducing kernel Hilbert space (RKHS) computed w.r.t. some distribution $\mathbb{P}$, has played a fundamental role as a basic building block of many kernel-based learning algorithms [2–4], and has recently gained increasing attention through the notion of embedding distributions in an RKHS [5–13]. Estimating the kernel mean remains an important problem as the underlying distribution $\mathbb{P}$ is usually unknown and we must rely entirely on the sample drawn according to $\mathbb{P}$.

Given a random sample drawn independently and identically (i.i.d.) from $\mathbb{P}$, the most common way to estimate the kernel mean is by replacing $\mathbb{P}$ by the empirical measure, $\mathbb{P}_n := \frac{1}{n} \sum_{i=1}^{n} \delta_{X_i}$ where $\delta_x$ is a Dirac measure at $x$ [5, 6]. Without any prior knowledge about $\mathbb{P}$, the empirical estimator is possibly the best one can do. However, [1] showed that this estimator can be "improved" by constructing a shrinkage estimator which is a combination of a model with low bias and high variance, and a model with high bias but low variance. Interestingly, significant improvement is in fact possible if the trade-off between these two models is chosen appropriately. The shrinkage estimator proposed in [1], which is motivated from the classical James-Stein shrinkage estimator [14] for the estimation of the mean of a normal distribution, is shown to have a smaller mean-squared error than that of the empirical estimator. These findings provide some support for the conceptual premise that we might be somewhat pessimistic in using the empirical estimator of the kernel mean and there is abundant room for further progress.

In this work, we adopt a spectral filtering approach to obtain shrinkage estimators of kernel mean that improve on the empirical estimator. The motivation behind our approach stems from the idea presented in [1] where the kernel mean estimation is reformulated as an empirical risk minimization (ERM) problem, with the shrinkage estimator being then obtained through penalized ERM. It is important to note that this motivation differs fundamentally from the typical supervised learning as the goal of regularization here is to get the James-Stein-like shrinkage estimators [14] rather than

to prevent overfitting. By looking at regularization from a filter function perspective, in this paper, we show that a wide class of shrinkage estimators for kernel mean can be obtained and that these estimators are consistent for an appropriate choice of the regularization/shrinkage parameter.

Unlike in earlier works [15–18] where the spectral filtering approach has been used in supervised learning problems, we here deal with unsupervised setting and only leverage spectral filtering as a way to construct a shrinkage estimator of the kernel mean. One of the advantages of this approach is that it allows us to incorporate meaningful prior knowledge. The resultant estimators are characterized by the filter function, which can be chosen according to the relevant prior knowledge. Moreover, the spectral filtering gives rise to a broader interpretation of shrinkage through, for example, the notion of early stopping and dimension reduction. Our estimators not only outperform the empirical estimator, but are also simple to implement and computationally efficient.

The paper is organized as follows. In Section 2, we introduce the problem of shrinkage estimation and present a new result that theoretically justifies the shrinkage estimator over the empirical estimator for kernel mean, which improves on the work of [1] while removing some of its drawbacks. Motivated by this result, we consider a general class of shrinkage estimators obtained via spectral filtering in Section 3 whose theoretical properties are presented in Section 4. The empirical performance of the proposed estimators are presented in Section 5. The missing proofs of the results are given in the supplementary material.

## 2 Kernel mean shrinkage estimator

In this section, we present preliminaries on the problem of shrinkage estimation in the context of estimating the kernel mean [1] and then present a theoretical justification (see Theorem 1) for shrinkage estimators that improves our understanding of the kernel mean estimation problem, while alleviating some of the issues inherent in the estimator proposed in [1].

**Preliminaries:** Let $\mathcal{H}$ be an RKHS of functions on a separable topological space $\mathcal{X}$. The space $\mathcal{H}$ is endowed with inner product $\langle \cdot, \cdot \rangle$, associated norm $\| \cdot \|$, and reproducing kernel $k : \mathcal{X} \times \mathcal{X} \to \mathbb{R}$, which we assume to be continuous and bounded, i.e., $\kappa := \sup_{x \in \mathcal{X}} \sqrt{k(x, x)} < \infty$. The kernel mean of some unknown distribution $\mathbb{P}$ on $\mathcal{X}$ and its empirical estimate—we refer to this as *kernel mean estimator* (KME)—from i.i.d. sample $x_1, \ldots, x_n$ are given by

$$\mu_{\mathbb{P}} := \int_{\mathcal{X}} k(x, \cdot) \, d\mathbb{P}(x) \qquad \text{and} \qquad \hat{\mu}_{\mathbb{P}} := \frac{1}{n} \sum_{i=1}^{n} k(x_i, \cdot), \qquad (1)$$

respectively. As mentioned before, $\hat{\mu}_{\mathbb{P}}$ is the "best" possible estimator to estimate $\mu_{\mathbb{P}}$ if nothing is known about $\mathbb{P}$. However, depending on the information that is available about $\mathbb{P}$, one can construct various estimators of $\mu_{\mathbb{P}}$ that perform "better" than $\mu_{\mathbb{P}}$. Usually, the performance measure that is used for comparison is the mean-squared error though alternate measures can be used. Therefore, our main objective is to improve upon KME in terms of the mean-squared error, i.e., construct $\tilde{\mu}_{\mathbb{P}}$ such that $\mathbb{E}_{\mathbb{P}} \| \tilde{\mu}_{\mathbb{P}} - \mu_{\mathbb{P}} \|^2 \leq \mathbb{E}_{\mathbb{P}} \| \hat{\mu}_{\mathbb{P}} - \mu_{\mathbb{P}} \|^2$ for all $\mathbb{P} \in \mathcal{P}$ with strict inequality holding for at least one element in $\mathcal{P}$ where $\mathcal{P}$ is a suitably large class of Borel probability measures on $\mathcal{X}$. Such an estimator $\tilde{\mu}_{\mathbb{P}}$ is said to be *admissible* w.r.t $\mathcal{P}$. If $\mathcal{P} = M_+^1(\mathcal{X})$ is the set of all Borel probability measures on $\mathcal{X}$, then $\tilde{\mu}_{\mathbb{P}}$ satisfying the above conditions may not exist and in that sense, $\hat{\mu}_{\mathbb{P}}$ is possibly the best estimator of $\mu_{\mathbb{P}}$ that one can have.

**Admissibility of shrinkage estimator:** To improve upon KME, motivated by the James-Stein estimator, $\tilde{\theta}$, [1] proposed a shrinkage estimator $\hat{\mu}_{\alpha} := \alpha f^* + (1 - \alpha) \hat{\mu}_{\mathbb{P}}$ where $\alpha \in \mathbb{R}$ is the shrinkage parameter that balances the low-bias, high-variance model ($\hat{\mu}_{\mathbb{P}}$) with the high-bias, low-variance model ($f^* \in \mathcal{H}$). Assuming for simplicity $f^* = 0$, [1] showed that $\mathbb{E}_{\mathbb{P}} \| \hat{\mu}_{\alpha} - \mu_{\mathbb{P}} \|^2 < \mathbb{E}_{\mathbb{P}} \| \hat{\mu}_{\mathbb{P}} - \mu_{\mathbb{P}} \|^2$ if and only if $\alpha \in (0, 2\Delta/(\Delta + \| \mu_{\mathbb{P}} \|^2))$ where $\Delta := \mathbb{E}_{\mathbb{P}} \| \hat{\mu}_{\mathbb{P}} - \mu_{\mathbb{P}} \|^2$. While this is an interesting result, the resultant estimator $\hat{\mu}_{\alpha}$ is strictly not a "statistical estimator" as it depends on quantities that need to be estimated, i.e., it depends on $\alpha$ whose choice requires the knowledge of $\mu_{\mathbb{P}}$, which is the quantity to be estimated. We would like to mention that [1] handles the general case with $f^*$ being not necessarily zero, wherein the range for $\alpha$ then depends on $f^*$ as well. But for the purposes of simplicity and ease of understanding, for the rest of this paper we assume $f^* = 0$. Since $\hat{\mu}_{\alpha}$ is not practically interesting, [1] resorted to the following representation of $\mu_{\mathbb{P}}$ and $\hat{\mu}_{\mathbb{P}}$ as solutions to the minimization problems [1, 19]:

$$\mu_{\mathbb{P}} = \arg \inf_{g \in \mathcal{H}} \int_{\mathcal{X}} \|k(x, \cdot) - g\|^2 \, d\mathbb{P}(x), \qquad \hat{\mu}_{\mathbb{P}} = \arg \inf_{g \in \mathcal{H}} \frac{1}{n} \sum_{i=1}^{n} \|k(x_i, \cdot) - g\|^2, \qquad (2)$$

using which $\hat{\mu}_{\alpha}$ is shown to be the solution to the regularized empirical risk minimization problem:

$$\check{\mu}_{\lambda} = \arg \inf_{g \in \mathcal{H}} \frac{1}{n} \sum_{i=1}^{n} \|k(x_i, \cdot) - g\|^2 + \lambda \|g\|^2, \qquad (3)$$

where $\lambda > 0$ and $\alpha := \frac{\lambda}{\lambda+1}$, i.e., $\check{\mu}_{\lambda} = \hat{\mu}_{\frac{\lambda}{\lambda+1}}$. It is interesting to note that unlike in supervised learning (e.g., least squares regression), the empirical minimization problem in (2) is not ill-posed and therefore does not require a regularization term although it is used in (3) to obtain a shrinkage estimator of $\mu_{\mathbb{P}}$. [1] then obtained a value for $\lambda$ through cross-validation and used it to construct $\hat{\mu}_{\frac{\lambda}{\lambda+1}}$ as an estimator of $\mu_{\mathbb{P}}$, which is then shown to perform empirically better than $\hat{\mu}_{\mathbb{P}}$. However, no theoretical guarantees including the basic requirement of $\hat{\mu}_{\frac{\lambda}{\lambda+1}}$ being consistent are provided. In fact, because $\lambda$ is data-dependent, the above mentioned result about the improved performance of $\hat{\mu}_{\alpha}$ over a range of $\alpha$ does not hold as such a result is proved assuming $\alpha$ is a constant and does not depend on the data. While it is clear that the regularizer in (3) is not needed to make (2) well-posed, the role of $\lambda$ is not clear from the point of view of $\hat{\mu}_{\frac{\lambda}{\lambda+1}}$ being consistent and better than $\hat{\mu}_{\mathbb{P}}$. The following result provides a theoretical understanding of $\hat{\mu}_{\frac{\lambda}{\lambda+1}}$ from these viewpoints.

**Theorem 1.** *Let $\check{\mu}_{\lambda}$ be constructed as in (3). Then the following hold.*

*(i)* $\|\check{\mu}_{\lambda} - \mu_{\mathbb{P}}\| \xrightarrow{\mathbb{P}} 0$ *as* $\lambda \to 0$ *and* $n \to \infty$. *In addition, if* $\lambda = n^{-\beta}$ *for some* $\beta > 0$, *then* $\|\check{\mu}_{\lambda} - \mu_{\mathbb{P}}\| = O_{\mathbb{P}}(n^{-\min\{\beta, 1/2\}})$.

*(ii) For* $\lambda = cn^{-\beta}$ *with* $c > 0$ *and* $\beta > 1$, *define* $\mathcal{P}_{c,\beta} := \{\mathbb{P} \in M_+^1(\mathcal{X}) : \|\mu_{\mathbb{P}}\|^2 < A \int k(x, x) \, d\mathbb{P}(x)\}$ *where* $A := \frac{2^{1/\beta}\beta}{2^{1/\beta}\beta + c^{1/\beta}(\beta-1)^{(\beta-1)/\beta}}$. *Then* $\forall n$ *and* $\forall \mathbb{P} \in \mathcal{P}_{c,\beta}$, *we have* $\mathbb{E}_{\mathbb{P}}\|\check{\mu}_{\lambda} - \mu_{\mathbb{P}}\|^2 < \mathbb{E}_{\mathbb{P}}\|\hat{\mu}_{\mathbb{P}} - \mu_{\mathbb{P}}\|^2$.

*Remark.* (*i*) Theorem 1(i) shows that $\check{\mu}_{\lambda}$ is a consistent estimator of $\mu_{\mathbb{P}}$ as long as $\lambda \to 0$ and the convergence rate in probability of $\|\check{\mu}_{\lambda} - \mu_{\mathbb{P}}\|$ is determined by the rate of convergence of $\lambda$ to zero, with the best possible convergence rate being $n^{-1/2}$. Therefore to attain a fast rate of convergence, it is instructive to choose $\lambda$ such that $\lambda\sqrt{n} \to 0$ as $\lambda \to 0$ and $n \to \infty$.

(*ii*) Suppose for some $c > 0$ and $\beta > 1$, we choose $\lambda = cn^{-\beta}$, which means the resultant estimator $\check{\mu}_{\lambda}$ is a proper estimator as it does not depend on any unknown quantities. Theorem 1(ii) shows that for any $n$ and $\mathbb{P} \in \mathcal{P}_{c,\beta}$, $\check{\mu}_{\lambda}$ is a "better" estimator than $\hat{\mu}_{\mathbb{P}}$. Note that for any $\mathbb{P} \in M_+^1(\mathcal{X})$, $\|\mu_{\mathbb{P}}\|^2 = \int \int k(x, y) \, d\mathbb{P}(x) \, d\mathbb{P}(y) \leq (\int \sqrt{k(x, x)} \, d\mathbb{P}(x))^2 \leq \int k(x, x) \, d\mathbb{P}(x)$. This means $\check{\mu}_{\lambda}$ is admissible if we restrict $M_+^1(\mathcal{X})$ to $\mathcal{P}_{c,\beta}$ which considers only those distributions for which $\|\mu_{\mathbb{P}}\|^2 / \int k(x, x) \, d\mathbb{P}(x)$ is strictly less than a constant, $A < 1$. It is obvious to note that if $c$ is very small or $\beta$ is very large, then $A$ gets closer to one and $\check{\mu}_{\lambda}$ behaves almost like $\hat{\mu}_{\mathbb{P}}$, thereby matching with our intuition.

(*iii*) A nice interpretation for $\mathcal{P}_{c,\beta}$ can be obtained as in Theorem 1(ii) when $k$ is a translation invariant kernel on $\mathbb{R}^d$. It can be shown that $\mathcal{P}_{c,\beta}$ contains the class of all probability measures whose characteristic function has an $L^2$ norm (and therefore is the set of square integrable probability densities if $\mathbb{P}$ has a density w.r.t. the Lebesgue measure) bounded by a constant that depends on $c$, $\beta$ and $k$ (see §2 in the supplementary material). ∎

## 3   Spectral kernel mean shrinkage estimator

Let us return to the shrinkage estimator $\hat{\mu}_{\alpha}$ considered in [1], i.e., $\hat{\mu}_{\alpha} = \alpha f^* + (1 - \alpha)\hat{\mu}_{\mathbb{P}} = \alpha \sum_i \langle f^*, e_i \rangle e_i + (1 - \alpha) \sum_i \langle \hat{\mu}_{\mathbb{P}}, e_i \rangle e_i$, where $(e_i)_{i \in \mathbb{N}}$ are the countable orthonormal basis (ONB) of $\mathcal{H}$—countable ONB exist since $\mathcal{H}$ is separable which follows from $\mathcal{X}$ being separable and $k$ being continuous [20, Lemma 4.33]. This estimator can be generalized by considering the shrinkage estimator $\hat{\mu}_{\boldsymbol{\alpha}} := \sum_i \alpha_i \langle f^*, e_i \rangle e_i + \sum_i (1 - \alpha_i) \langle \hat{\mu}_{\mathbb{P}}, e_i \rangle e_i$ where $\boldsymbol{\alpha} := (\alpha_1, \alpha_2, \ldots) \in \mathbb{R}^\infty$ is a sequence of shrinkage parameters. If $\Delta_{\boldsymbol{\alpha}} := \mathbb{E}_{\mathbb{P}}\|\hat{\mu}_{\boldsymbol{\alpha}} - \mu_{\mathbb{P}}\|^2$ is the risk of this estimator, the following theorem gives an optimality condition on $\boldsymbol{\alpha}$ for which $\Delta_{\boldsymbol{\alpha}} < \Delta$.

**Theorem 2.** *For some ONB $(e_i)_i$, $\Delta_{\boldsymbol{\alpha}} - \Delta = \sum_i (\Delta_{\boldsymbol{\alpha},i} - \Delta_i)$ where $\Delta_{\boldsymbol{\alpha},i}$ and $\Delta_i$ denote the risk of the ith component of $\hat{\mu}_{\boldsymbol{\alpha}}$ and $\hat{\mu}_{\mathbb{P}}$, respectively. Then, $\Delta_{\boldsymbol{\alpha},i} - \Delta_i < 0$ if*

$$0 < \alpha_i < \frac{2\Delta_i}{\Delta_i + (f_i^* - \mu_i)^2}, \qquad (4)$$

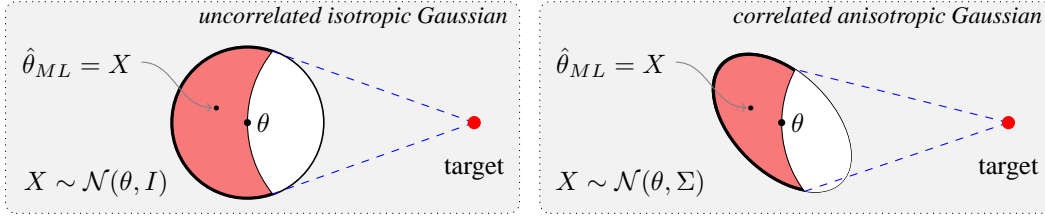

Figure 1: Geometric explanation of a shrinkage estimator when estimating a mean of a Gaussian distribution. For isotropic Gaussian, the level sets of the joint density of $\hat{\theta}_{ML} = X$ are hyperspheres. In this case, shrinkage has the same effect regardless of the direction. Shaded area represents those estimates that get closer to $\theta$ after shrinkage. For anisotropic Gaussian, the level sets are concentric ellipsoids, which makes the effect dependent on the direction of shrinkage.

*where $f_i^*$ and $\mu_i$ denote the Fourier coefficients of $f^*$ and $\mu_{\mathbb{P}}$, respectively.*

The condition in (4) is a component-wise version of the condition given in [1, Theorem 1] for a class of estimators $\hat{\mu}_\alpha := \alpha f^* + (1 - \alpha)\hat{\mu}_{\mathbb{P}}$ which may be expressed here by assuming that we have a constant shrinkage parameter $\alpha_i = \alpha$ for all $i$. Clearly, as the optimal range of $\alpha_i$ may vary across coordinates, the class of estimators in [1] does not allow us to adjust $\alpha_i$ accordingly. To understand why this property is important, let us consider the problem of estimating the mean of Gaussian distribution illustrated in Figure 1. For correlated random variable $X \sim \mathcal{N}(\theta, \Sigma)$, a natural choice of basis is the set of orthonormal eigenvectors which diagonalize the covariance matrix $\Sigma$ of $X$. Clearly, the optimal range of $\alpha_i$ depends on the corresponding eigenvalues. Allowing for different basis $(e_i)_i$ and shrinkage parameter $\alpha_i$ opens up a wide range of strategies that can be used to construct "better" estimators.

A natural strategy under this representation is as follows: *i*) we specify the ONB $(e_i)_i$ and project $\hat{\mu}_{\mathbb{P}}$ onto this basis. *ii*) we shrink each $\hat{\mu}_i$ independently according to a pre-defined shrinkage rule. *iii*) the shrinkage estimate is reconstructed as a superposition of the resulting components. In other words, an ideal shrinkage estimator can be defined formally as a non-linear mapping:

$$\hat{\mu}_{\mathbb{P}} \longrightarrow \sum_i h(\alpha_i)\langle f^*, e_i \rangle e_i + \sum_i (1 - h(\alpha_i))\langle \hat{\mu}_{\mathbb{P}}, e_i \rangle e_i \tag{5}$$

where $h : \mathbb{R} \to \mathbb{R}$ is a shrinkage rule. Since we make no reference to any particular basis $(e_i)_i$, nor to any particular shrinkage rule $h$, a wide range of strategies can be adopted here. For example, we can view *whitening* as a special case in which $f^*$ is the data average $\frac{1}{n}\sum_{i=1}^n x_i$ and $1 - h(\alpha_i) = 1/\sqrt{\alpha_i}$ where $\alpha_i$ and $e_i$ are the $i$th eigenvalue and eigenvector of the covariance matrix, respectively.

Inspired by Theorem 2, we adopt the spectral filtering approach as one of the strategies to construct the estimators of the form (5). To this end, owing to the regularization interpretation in (3), we consider estimators of the form $\sum_{i=1}^n \beta_i k(x_i, \cdot)$ for some $\boldsymbol{\beta} \in \mathbb{R}^n$—looking for such an estimator is equivalent to learning a *signed measure* that is supported on $(x_i)_{i=1}^n$. Since $\sum_{i=1}^n \beta_i k(x_i, \cdot)$ is a minimizer of (3), $\boldsymbol{\beta}$ should satisfy $\mathbf{K}\boldsymbol{\beta} = \mathbf{K}\mathbf{1}_n$ where $\mathbf{K}$ is an $n \times n$ Gram matrix and $\mathbf{1}_n = [1/n, \dots, 1/n]^\top$. Here the solution is trivially $\boldsymbol{\beta} = \mathbf{1}_n$, i.e., the coefficients of the standard estimator $\hat{\mu}_{\mathbb{P}}$ if $\mathbf{K}$ is invertible. Since $\mathbf{K}^{-1}$ may not exist and even if it exists, the computation of it can be numerically unstable, the idea of spectral filtering—this is quite popular in the theory of inverse problems [15] and has been used in kernel least squares [17]—is to replace $\mathbf{K}^{-1}$ by some regularized matrices $g_\lambda(\mathbf{K})$ that approximates $\mathbf{K}^{-1}$ as $\lambda$ goes to zero. Note that unlike in (3), the regularization is quite important here (i.e., the case of estimators of the form $\sum_{i=1}^n \beta_i k(x_i, \cdot)$) without which the the linear system is under determined. Therefore, we propose the following class of estimators:

$$\hat{\mu}_\lambda := \sum_{i=1}^n \beta_i k(x_i, \cdot) \quad \text{with} \quad \boldsymbol{\beta}(\lambda) := g_\lambda(\mathbf{K})\mathbf{K}\mathbf{1}_n, \tag{6}$$

where $g_\lambda(\cdot)$ is a filter function and $\lambda$ is referred to as a shrinkage parameter. The matrix-valued function $g_\lambda(\mathbf{K})$ can be described by a scalar function $g_\lambda : [0, \kappa^2] \to \mathbb{R}$ on the spectrum of $\mathbf{K}$. That is, if $\mathbf{K} = \mathbf{U}\mathbf{D}\mathbf{U}^\top$ is the eigen-decomposition of $\mathbf{K}$ where $\mathbf{D} = \text{diag}(\tilde{\gamma}_1, \dots, \tilde{\gamma}_n)$, we have $g_\lambda(\mathbf{D}) = \text{diag}(g_\lambda(\tilde{\gamma}_1), \dots, g_\lambda(\tilde{\gamma}_n))$ and $g_\lambda(\mathbf{K}) = \mathbf{U}g_\lambda(\mathbf{D})\mathbf{U}^\top$. For example, the scalar filter function of Tikhonov regularization is $g_\lambda(\gamma) = 1/(\gamma + \lambda)$. In the sequel, we call this class of estimators a *spectral kernel mean shrinkage estimator* (Spectral-KMSE).

Table 1: Update equations for $\boldsymbol{\beta}$ and corresponding filter functions.

| Algorithm | Update Equation ($\mathbf{a} := \mathbf{K1}_n - \mathbf{K}\boldsymbol{\beta}^{t-1}$) | Filter Function |
|---|---|---|
| L2 Boosting | $\boldsymbol{\beta}^t \leftarrow \boldsymbol{\beta}^{t-1} + \eta\mathbf{a}$ | $g(\gamma) = \eta\sum_{i=1}^{t-1}(1-\eta\gamma)^i$ |
| Acc. L2 Boosting | $\boldsymbol{\beta}^t \leftarrow \boldsymbol{\beta}^{t-1} + \omega_t(\boldsymbol{\beta}^{t-1} - \boldsymbol{\beta}^{t-2}) + \frac{\kappa_t}{n}\mathbf{a}$ | $g(\gamma) = p_t(\gamma)$ |
| Iterated Tikhonov | $(\mathbf{K} + n\lambda\mathbf{I})\boldsymbol{\beta}_i = \mathbf{1}_n + n\lambda\boldsymbol{\beta}_{i-1}$ | $g(\gamma) = \frac{(\gamma+\lambda)^t - \gamma^t}{\lambda(\gamma+\lambda)^t}$ |
| Truncated SVD | None | $g(\gamma) = \gamma^{-1}\mathbb{1}_{\{\gamma \geq \lambda\}}$ |

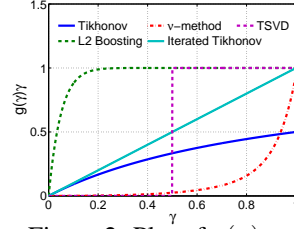

Figure 2: Plot of $g(\gamma)\gamma$.

**Proposition 3.** *The Spectral-KMSE satisfies* $\hat{\mu}_\lambda = \sum_{i=1}^n g_\lambda(\tilde{\gamma}_i)\tilde{\gamma}_i\langle\hat{\mu}, \tilde{\mathbf{v}}_i\rangle\tilde{\mathbf{v}}_i$, *where* $(\tilde{\gamma}_i, \tilde{\mathbf{v}}_i)$ *are eigenvalue and eigenfunction pairs of the empirical covariance operator* $\widehat{\mathcal{C}}_k : \mathcal{H} \to \mathcal{H}$ *defined as* $\widehat{\mathcal{C}}_k = \frac{1}{n}\sum_{i=1}^n k(\cdot, x_i) \otimes k(\cdot, x_i)$.

By virtue of Proposition 3, if we choose $1 - h(\tilde{\gamma}) := g_\lambda(\tilde{\gamma})\tilde{\gamma}$, the Spectral-KMSE is indeed in the form of (5) when $f^* = 0$ and $(e_i)_i$ is the kernel PCA (KPCA) basis, with the filter function $g_\lambda$ determining the shrinkage rule. Since by definition $g_\lambda(\tilde{\gamma}_i)$ approaches the function $1/\tilde{\gamma}_i$ as $\lambda$ goes to 0, the function $g_\lambda(\tilde{\gamma}_i)\tilde{\gamma}_i$ approaches 1 (no shrinkage). As the value of $\lambda$ increases, we have more shrinkage because the value of $g_\lambda(\tilde{\gamma}_i)\tilde{\gamma}_i$ deviates from 1, and the behavior of this deviation depends on the filter function $g_\lambda$. For example, we can see that Proposition 3 generalizes Theorem 2 in [1] where the filter function is $g_\lambda(\mathbf{K}) = (\mathbf{K} + n\lambda\mathbf{I})^{-1}$, i.e., $g(\gamma) = 1/(\gamma + \lambda)$. That is, we have $g_\lambda(\tilde{\gamma}_i)\tilde{\gamma}_i = \tilde{\gamma}_i/(\tilde{\gamma}_i + \lambda)$, implying that the effect of shrinkage is relatively larger in the low-variance direction. In the following, we discuss well-known examples of spectral filtering algorithms obtained by various choices of $g_\lambda$. Update equations for $\boldsymbol{\beta}(\lambda)$ and corresponding filter functions are summarized in Table 1. Figure 2 illustrates the behavior of these filter functions.

**L2 Boosting.** This algorithm, also known as gradient descent or Landweber iteration, finds a weight $\boldsymbol{\beta}$ by performing a gradient descent iteratively. Thus, we can interpret *early stopping* as shrinkage and the reciprocal of iteration number as shrinkage parameter, i.e., $\lambda \approx 1/t$. The step-size $\eta$ does not play any role for shrinkage [16], so we use the fixed step-size $\eta = 1/\kappa^2$ throughout.

**Accelerated L2 Boosting.** This algorithm, also known as $\nu$-method, uses an accelerated gradient descent step, which is faster than L2 Boosting because we only need $\sqrt{t}$ iterations to get the same solution as the L2 Boosting would get after $t$ iterations. Consequently, we have $\lambda \approx 1/t^2$.

**Iterated Tikhonov.** This algorithm can be viewed as a combination of Tikhonov regularization and gradient descent. Both parameters $\lambda$ and $t$ play the role of shrinkage parameter.

**Truncated Singular Value Decomposition.** This algorithm can be interpreted as a projection onto the first principal components of the KPCA basis. Hence, we may interpret *dimensionality reduction* as shrinkage and the size of reduced dimension as shrinkage parameter. This approach has been used in [21] to improve the kernel mean estimation under the low-rank assumption.

Most of the above spectral filtering algorithms allow to compute the coefficients $\boldsymbol{\beta}$ without explicitly computing the eigen-decomposition of $\mathbf{K}$, as we can see in Table 1, and some of which may have no natural interpretation in terms of regularized risk minimization. Lastly, an initialization of $\boldsymbol{\beta}$ corresponds to the target of shrinkage. In this work, we assume that $\boldsymbol{\beta}^0 = 0$ throughout.

## 4 Theoretical properties of Spectral-KMSE

This section presents some theoretical properties for the proposed Spectral-KMSE in (6). To this end, we first present a regularization interpretation that is different from the one in (3) which involves learning a smooth operator from $\mathcal{H}$ to $\mathcal{H}$ [22]. This will be helpful to investigate the consistency of the Spectral-KMSE. Let us consider the following regularized risk minimization problem,

$$\arg\min_{\mathbf{F}\in\mathcal{H}\otimes\mathcal{H}} \quad \mathbb{E}_X \|k(X, \cdot) - \mathbf{F}[k(X, \cdot)]\|_{\mathcal{H}}^2 + \lambda\|\mathbf{F}\|_{HS}^2 \tag{7}$$

where $\mathbf{F}$ is a Hilbert-Schmidt operator from $\mathcal{H}$ to $\mathcal{H}$. Essentially, we are seeking a smooth operator $\mathbf{F}$ that maps $k(x, \cdot)$ to itself, where (7) is an instance of the regression framework in [22]. The formulation of shrinkage as the solution of a smooth operator regression, and the empirical solution (8) and in the lines below, were given in a personal communication by Arthur Gretton. It can be

shown that the solution to (7) is given by $\mathbf{F} = \mathcal{C}_k(\mathcal{C}_k + \lambda\mathbf{I})^{-1}$ where $\mathcal{C}_k : \mathcal{H} \to \mathcal{H}$ is a covariance operator in $\mathcal{H}$ defined as $\mathcal{C}_k = \int k(\cdot, x) \otimes k(\cdot, x) \, d\mathbb{P}(x)$ (see §5 of the supplement for a proof). Define $\mu_\lambda := \mathbf{F}\mu_\mathbb{P} = \mathcal{C}_k(\mathcal{C}_k + \lambda\mathbf{I})^{-1}\mu_\mathbb{P}$. Since $k$ is bounded, it is easy to verify that $\mathcal{C}_k$ is Hilbert-Schmidt and therefore compact. Hence by the Hilbert-Schmidt theorem, $\mathcal{C}_k = \sum_i \gamma_i \langle \cdot, \psi_i \rangle \psi_i$ where $(\gamma_i)_{i\in\mathbb{N}}$ are the positive eigenvalues and $(\psi_i)_{i\in\mathbb{N}}$ are the corresponding eigenvectors that form an ONB for the range space of $\mathcal{C}_k$ denoted as $\mathcal{R}(\mathcal{C}_k)$. This implies $\mu_\lambda$ can be decomposed as $\mu_\lambda = \sum_{i=1}^\infty \frac{\gamma_i}{\gamma_i + \lambda} \langle \mu_\mathbb{P}, \psi_i \rangle \psi_i$. We can observe that the filter function corresponding to the problem (7) is $g_\lambda(\gamma) = 1/(\gamma + \lambda)$. By extending this approach to other filter functions, we obtain $\mu_\lambda = \sum_{i=1}^\infty \gamma_i g_\lambda(\gamma_i) \langle \mu_\mathbb{P}, \psi_i \rangle \psi_i$ which is equivalent to $\mu_\lambda = \mathcal{C}_k g_\lambda(\mathcal{C}_k)\mu_\mathbb{P}$.

Since $\mathcal{C}_k$ is a compact operator, the role of filter function $g_\lambda$ is to regularize the inverse of $\mathcal{C}_k$. In standard supervised setting, the explicit form of the solution is $f_\lambda = g_\lambda(L_k)L_k f_\rho$ where $L_k$ is the integral operator of kernel $k$ acting in $L^2(\mathcal{X}, \rho_X)$ and $f_\rho$ is the expected solution given by $f_\rho(x) = \int_\mathcal{Y} y \, d\rho(y|x)$ [16]. It is interesting to see that $\mu_\lambda$ admits a similar form to that of $f_\lambda$, but it is written in term of covariance operator $\mathcal{C}_k$ instead of the integral operator $L_k$. Moreover, the solution to (7) is also in a similar form to the regularized conditional embedding $\mu_{Y|X} = \mathcal{C}_{YX}(\mathcal{C}_k + \lambda\mathbf{I})^{-1}$ [9]. This connection implies that the spectral filtering may be applied more broadly to improve the estimation of conditional mean embedding, i.e., $\mu_{Y|X} = \mathcal{C}_{YX}g_\lambda(\mathcal{C}_k)$.

The empirical counterpart of (7) is given by

$$\arg\min_{\mathbf{F}} \quad \frac{1}{n}\sum_{i=1}^n \|k(x_i, \cdot) - \mathbf{F}[k(x_i, \cdot)]\|_\mathcal{H}^2 + \lambda\|\mathbf{F}\|_{HS}^2, \tag{8}$$

resulting in $\hat{\mu}_\lambda = \mathbf{F}\hat{\mu}_\mathbb{P} = \mathbf{1}_n^\top \mathbf{K}(\mathbf{K} + \lambda\mathbf{I})^{-1}\Phi$ where $\Phi = [k(x_1, \cdot), \dots, k(x_n, \cdot)]^\top$, which matches with the one in (6) with $g_\lambda(\mathbf{K}) = (\mathbf{K} + \lambda\mathbf{I})^{-1}$. Note that this is exactly the F-KMSE proposed in [1]. Based on $\mu_\lambda$ which depends on $\mathbb{P}$, an empirical version of it can be obtained by replacing $\mathcal{C}_k$ and $\mu_\mathbb{P}$ with their empirical estimators leading to $\tilde{\mu}_\lambda = \hat{\mathcal{C}}_k g_\lambda(\hat{\mathcal{C}}_k)\hat{\mu}_\mathbb{P}$. The following result shows that $\hat{\mu}_\lambda = \tilde{\mu}_\lambda$, which means the Spectral-KMSE proposed in (6) is equivalent to solving (8).

**Proposition 4.** *Let $\hat{\mathcal{C}}_k$ and $\hat{\mu}_\mathbb{P}$ be the sample counterparts of $\mathcal{C}_k$ and $\mu_\mathbb{P}$ given by $\hat{\mathcal{C}}_k := \frac{1}{n}\sum_{i=1}^n k(x_i, \cdot) \otimes k(x_i, \cdot)$ and $\hat{\mu}_\mathbb{P} := \frac{1}{n}\sum_{i=1}^n k(x_i, \cdot)$, respectively. Then, we have that $\tilde{\mu}_\lambda := \hat{\mathcal{C}}_k g_\lambda(\hat{\mathcal{C}}_k)\hat{\mu}_\mathbb{P} = \hat{\mu}_\lambda$, where $\hat{\mu}_\lambda$ is defined in (6).*

Having established a regularization interpretation for $\hat{\mu}_\lambda$, it is of interest to study the consistency and convergence rate of $\hat{\mu}_\lambda$ similar to KMSE in Theorem 1. Our main goal here is to derive convergence rates for a broad class of algorithms given a set of sufficient conditions on the filter function, $g_\lambda$. We believe that for some algorithms it is possible to derive the best achievable bounds, which requires ad-hoc proofs for each algorithm. To this end, we provide a set of conditions any *admissible* filter function, $g_\lambda$ must satisfy.

**Definition 1.** *A family of filter functions $g_\lambda : [0, \kappa^2] \to \mathbb{R}, 0 < \lambda \leq \kappa^2$ is said to be admissible if there exists finite positive constants $B$, $C$, $D$, and $\eta_0$ (all independent of $\lambda$) such that $(C1) \sup_{\gamma\in[0,\kappa^2]} |\gamma g_\lambda(\gamma)| \leq B$, $(C2) \sup_{\gamma\in[0,\kappa^2]} |r_\lambda(\gamma)| \leq C$ and $(C3) \sup_{\gamma\in[0,\kappa^2]} |r_\lambda(\gamma)|\gamma^\eta \leq D\lambda^\eta$, $\forall \eta \in (0, \eta_0]$ hold, where $r_\lambda(\gamma) := 1 - \gamma g_\lambda(\gamma)$.*

These conditions are quite standard in the theory of inverse problems [15, 23]. The constant $\eta_0$ is called the *qualification* of $g_\lambda$ and is a crucial factor that determines the rate of convergence in inverse problems. As we will see below, that the rate of convergence of $\hat{\mu}_\lambda$ depends on two factors: (a) smoothness of $\mu_\mathbb{P}$ which is usually unknown as it depends on the unknown $\mathbb{P}$ and (b) qualification of $g_\lambda$ which determines how well the smoothness of $\mu_\mathbb{P}$ is captured by the spectral filter, $g_\lambda$.

**Theorem 5.** *Suppose $g_\lambda$ is admissible in the sense of Definition 1. Let $\kappa = \sup_{x\in\mathcal{X}} \sqrt{k(x,x)}$. If $\mu_\mathbb{P} \in \mathcal{R}(\mathcal{C}_k^\beta)$ for some $\beta > 0$, then for any $\delta > 0$, with probability at least $1 - 3e^{-\delta}$,*

$$\|\hat{\mu}_\lambda - \mu_\mathbb{P}\| \leq \frac{2\kappa B + \kappa B\sqrt{2\delta}}{\sqrt{n}} + D\lambda^{\min\{\beta,\eta_0\}}\|\mathcal{C}_k^{-\beta}\mu_\mathbb{P}\| + C\tau\frac{(2\sqrt{2}\kappa^2\sqrt{\delta})^{\min\{1,\beta\}}}{n^{\min\{1/2,\beta/2\}}}\|\mathcal{C}_k^{-\beta}\mu_\mathbb{P}\|,$$

*where $\mathcal{R}(A)$ denotes the range space of $A$ and $\tau$ is some universal constant that does not depend on $\lambda$ and $n$. Therefore, $\|\hat{\mu}_\lambda - \mu_\mathbb{P}\| = O_\mathbb{P}(n^{-\min\{1/2,\beta/2\}})$ with $\lambda = o(n^{-\frac{\min\{1/2,\beta/2\}}{\min\{\beta,\eta_0\}}})$.*

Theorem 5 shows that the convergence rate depends on the smoothness of $\mu_\mathbb{P}$ which is imposed through the range space condition that $\mu_\mathbb{P} \in \mathcal{R}(\mathcal{C}_k^\beta)$ for some $\beta > 0$. Note that this is in contrast

to the estimator in Theorem 1 which does not require any smoothness assumptions on $\mu_{\mathbb{P}}$. It can be shown that the smoothness of $\mu_{\mathbb{P}}$ increases with increase in $\beta$. This means, irrespective of the smoothness of $\mu_{\mathbb{P}}$ for $\beta > 1$, the best possible convergence rate is $n^{-1/2}$ which matches with that of KMSE in Theorem 1. While the qualification $\eta_0$ does not seem to directly affect the rates, it controls the rate at which $\lambda$ converges to zero. For example, if $g_\lambda(\gamma) = 1/(\gamma + \lambda)$ which corresponds to Tikhonov regularization, it can be shown that $\eta_0 = 1$ which means for $\beta > 1$, $\lambda = o(n^{-1/2})$ implying that $\lambda$ cannot decay to zero slower than $n^{-1/2}$. Ideally, one would require a larger $\eta_0$ (preferably infinity which is the case with truncated SVD) so that the convergence of $\lambda$ to zero can be made arbitrarily slow if $\beta$ is large. This way, both $\beta$ and $\eta_0$ control the behavior of the estimator.

In fact, Theorem 5 provides a choice for $\lambda$—which is what we used in Theorem 1 to study the admissibility of $\breve{\mu}_\lambda$ to $\mathcal{P}_{c,\beta}$—to construct the Spectral-KMSE. However, this choice of $\lambda$ depends on $\beta$ which is not known in practice (although $\eta_0$ is known as it is determined by the choice of $g_\lambda$). Therefore, $\lambda$ is usually learnt from data through cross-validation or through Lepski's method [24] for which guarantees similar to the one presented in Theorem 5 can be provided. However, irrespective of the data-dependent/independent choice for $\lambda$, checking for the admissibility of Spectral-KMSE (similar to the one in Theorem 1) is very difficult and we intend to consider it in future work.

## 5 Empirical studies

**Synthetic data.** Given the i.i.d. sample $\mathbf{X} = \{x_1, x_2, \ldots, x_n\}$ from $\mathbb{P}$ where $x_i \in \mathbb{R}^d$, we evaluate different estimators using the loss function $L(\boldsymbol{\beta}, \mathbf{X}, \mathbb{P}) := \|\sum_{i=1}^n \beta_i k(x_i, \cdot) - \mathbb{E}_{x \sim \mathbb{P}}[k(x, \cdot)]\|_{\mathcal{H}}^2$. The risk of the estimator is subsequently approximated by averaging over $m$ independent copies of $\mathbf{X}$. In this experiment, we set $n = 50$, $d = 20$, and $m = 1000$. Throughout, we use the Gaussian RBF kernel $k(x, x') = \exp(-\|x - x'\|^2/2\sigma^2)$ whose bandwidth parameter is calculated using the median heuristic, i.e., $\sigma^2 = \text{median}\{\|x_i - x_j\|^2\}$. To allow for an analytic calculation of the loss $L(\boldsymbol{\beta}, \mathbf{X}, \mathbb{P})$, we assume that the distribution $\mathbb{P}$ is a $d$-dimensional mixture of Gaussians [1, 8]. Specifically, the data are generated as follows: $x \sim \sum_{i=1}^4 \pi_i \mathcal{N}(\boldsymbol{\theta}_i, \Sigma_i) + \varepsilon, \theta_{ij} \sim \mathcal{U}(-10, 10), \Sigma_i \sim \mathcal{W}(3 \times \mathbf{I}_d, 7), \varepsilon \sim \mathcal{N}(0, 0.2 \times \mathbf{I}_d)$ where $\mathcal{U}(a, b)$ and $\mathcal{W}(\Sigma_0, df)$ are the uniform distribution and Wishart distribution, respectively. As in [1], we set $\boldsymbol{\pi} = [0.05, 0.3, 0.4, 0.25]$.

A natural approach for choosing $\lambda$ is cross-validation procedure, which can be performed efficiently for the iterative methods such as Landweber and accelerated Landweber. For these two algorithms, we evaluate the leave-one-out score and select $\boldsymbol{\beta}^t$ at the iteration $t$ that minimizes this score (see, e.g., Figure 3(a)). Note that these methods have the built-in property of computing the whole *regularization path* efficiently. Since each iteration of the iterated Tikhonov is in fact equivalent to the F-KMSE, we assume $t = 3$ for simplicity and use the efficient LOOCV procedure proposed in [1] to find $\lambda$ at each iteration. Lastly, the truncation limit of TSVD can be identified efficiently by mean of generalized cross-validation (GCV) procedure [25]. To allow for an efficient calculation of GCV score, we resort to the alternative loss function $\mathcal{L}(\boldsymbol{\beta}) := \|\mathbf{K}\boldsymbol{\beta} - \mathbf{K}\mathbf{1}_n\|_2^2$.

Figure 3 reveals interesting aspects of the Spectral-KMSE. Firstly, as we can see in Figure 3(a), the number of iterations acts as shrinkage parameter whose optimal value can be attained within just a few iterations. Moreover, these methods do not suffer from "over-shrinking" because $\lambda \to 0$ as $t \to \infty$. In other words, if the chosen $t$ happens to be too large, the worst we can get is the standard empirical estimator. Secondly, Figure 3(b) demonstrates that both Landweber and accelerated Landweber are more computationally efficient than the F-KMSE. Lastly, Figure 3(c) suggests that the improvement of shrinkage estimators becomes increasingly remarkable in a high-dimensional setting. Interestingly, we can observe that most Spectral-KMSE algorithms outperform the S-KMSE, which supports our hypothesis on the importance of the geometric information of RKHS mentioned in Section 3. In addition, although the TSVD still gain from shrinkage, the improvement is smaller than other algorithms. This highlights the importance of filter functions and associated parameters.

**Real data.** We apply Spectral-KMSE to the density estimation problem via kernel mean matching [1, 26]. The datasets were taken from the UCI repository[1] and pre-processed by standardizing each feature. Then, we fit a mixture model $Q = \sum_{j=1}^r \pi_j \mathcal{N}(\boldsymbol{\theta}_j, \sigma_j^2 \mathbf{I})$ to the pre-processed dataset

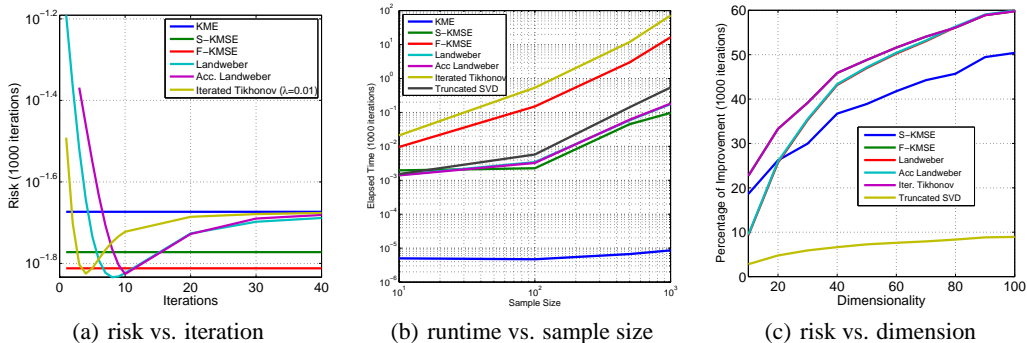

|  | (a) risk vs. iteration | (b) runtime vs. sample size | (c) risk vs. dimension |

Figure 3: (a) For iterative algorithms, the number of iterations acts as shrinkage parameter. (b) The iterative algorithms such as Landweber and accelerated Landweber are more efficient than the F-KMSE. (c) A percentage of improvement w.r.t. the KME, i.e., $100 \times (R - R_\lambda)/R$ where $R$ and $R_\lambda$ denote the approximated risk of KME and KMSE, respectively. Most Spectral-KMSE algorithms outperform S-KMSE which does not take into account the geometric information of the RKHS.

$\mathbf{X} := \{x_i\}_{i=1}^n$ by minimizing $\|\mu_Q - \hat{\mu}_X\|^2$ subject to the constraint $\sum_{j=1}^r \pi_j = 1$. Here $\mu_Q$ is the mean embedding of the mixture model $Q$ and $\hat{\mu}_X$ is the empirical mean embedding obtained from $\mathbf{X}$. Based on different estimators of $\mu_X$, we evaluate the resultant model $Q$ by the negative log-likelihood score on the test data. The parameters $(\pi_j, \boldsymbol{\theta}_j, \sigma_j^2)$ are initialized by the best one obtained from the $K$-means algorithm with 50 initializations. Throughout, we set $r = 5$ and use 25% of each dataset as a test set.

Table 2: The average negative log-likelihood evaluated on the test set. The results are obtained from 30 repetitions of the experiment. The boldface represents the statistically significant results.

| Dataset | KME | S-KMSE | F-KMSE | Landweber | Acc Land | Iter Tik | TSVD |
|---|---|---|---|---|---|---|---|
| ionosphere | 36.1769 | 36.1402 | 36.1622 | **36.1204** | **36.1554** | 36.1334 | 36.1442 |
| glass | 10.7855 | 10.7403 | 10.7448 | **10.7099** | 10.7541 | 10.9078 | 10.7791 |
| bodyfat | 18.1964 | 18.1158 | 18.1810 | 18.1607 | 18.1941 | 18.1267 | 18.1061 |
| housing | 14.3016 | **14.2195** | 14.0409 | 14.2499 | 14.1983 | **14.2868** | **14.3129** |
| vowel | 13.9253 | 13.8426 | 13.8817 | 13.8337 | 14.1368 | 13.8633 | 13.8375 |
| svmguide2 | 28.1091 | **28.0546** | 27.9640 | 28.1052 | **27.9693** | **28.0417** | 28.1128 |
| vehicle | 18.5295 | 18.3693 | **18.2547** | 18.4873 | **18.3124** | 18.4128 | 18.3910 |
| wine | 16.7668 | **16.7548** | 16.7457 | **16.7596** | 16.6790 | 16.6954 | **16.5719** |
| wdbc | 35.1916 | **35.1814** | 35.0023 | **35.1402** | 35.1366 | 35.1881 | **35.1850** |

Table 2 reports the results on real data. In general, the mixture model $Q$ obtained from the proposed shrinkage estimators tend to achieve lower negative log-likelihood score than that obtained from the standard empirical estimator. Moreover, we can observe that the relative performance of different filter functions vary across datasets, suggesting that, in addition to potential gain from shrinkage, incorporating prior knowledge through the choice of filter function could lead to further improvement.

## 6 Conclusion

We shows that several shrinkage strategies can be adopted to improve the kernel mean estimation. This paper considers the spectral filtering approach as one of such strategies. Compared to previous work [1], our estimators take into account the specifics of kernel methods and meaningful prior knowledge through the choice of filter functions, resulting in a wider class of shrinkage estimators. The theoretical analysis also reveals a fundamental similarity to standard supervised setting. Our estimators are simple to implement and work well in practice, as evidenced by the empirical results.

**Acknowledgments**

The first author thanks Ingo Steinwart for pointing out existing works along the line of spectral filtering, and Arthur Gretton for suggesting the connection of shrinkage to smooth operator framework. This work was carried out when the second author was a Research Fellow in the Statistical Laboratory, Department of Pure Mathematics and Mathematical Statistics at the University of Cambridge.

## Footnotes

[1] http://archive.ics.uci.edu/ml/

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
