[Supplementary Material]

# Kernel Mean Estimation via Spectral Filtering: Supplementary Material

**Krikamol Muandet**
MPI-IS, Tübingen
krikamol@tue.mpg.de

**Bharath Sriperumbudur**
Dep. of Statistics, PSU
bks18@psu.edu

**Bernhard Schölkopf**
MPI-IS, Tübingen
bs@tue.mpg.de

## Abstract

This note contains supplementary materials to *Kernel Mean Estimation via Spectral Filtering*.

## 1 Proof of Theorem 1

(i) Since $\check{\mu}_\lambda = \hat{\mu}_{\frac{\lambda}{\lambda+1}} = \frac{\hat{\mu}_\mathbb{P}}{\lambda+1}$, we have

$$\|\check{\mu}_\lambda - \mu_\mathbb{P}\| = \left\| \frac{\hat{\mu}_\mathbb{P}}{\lambda+1} - \mu_\mathbb{P} \right\| \leq \left\| \frac{\hat{\mu}_\mathbb{P}}{\lambda+1} - \frac{\mu_\mathbb{P}}{\lambda+1} \right\| + \left\| \frac{\mu_\mathbb{P}}{\lambda+1} - \mu_\mathbb{P} \right\| \leq \|\hat{\mu}_\mathbb{P} - \mu_\mathbb{P}\| + \lambda\|\mu_\mathbb{P}\|.$$

From [1], we have that $\|\hat{\mu}_\mathbb{P} - \mu_\mathbb{P}\| = O_\mathbb{P}(n^{-1/2})$ and therefore the result follows.

(ii) Define $\Delta := \mathbb{E}_\mathbb{P}\|\hat{\mu}_\mathbb{P} - \mu_\mathbb{P}\|^2 = \frac{\int k(x,x)\,\mathrm{d}\mathbb{P}(x) - \|\mu_\mathbb{P}\|^2}{n}$. Consider

$$
\begin{aligned}
\mathbb{E}_\mathbb{P}\|\check{\mu}_\lambda - \mu_\mathbb{P}\|^2 - \Delta &= \mathbb{E}_\mathbb{P}\left\| \frac{n^\beta}{n^\beta + c}(\hat{\mu}_\mathbb{P} - \mu_\mathbb{P}) - \mu_\mathbb{P} \right\|^2 - \Delta \\
&= \left( \frac{n^\beta}{n^\beta + c} \right)^2 \Delta + \frac{c^2}{(n^\beta + c)^2}\|\mu_\mathbb{P}\|^2 - \Delta \\
&= \frac{c^2\|\mu_\mathbb{P}\|^2 - (c^2 + 2cn^\beta)\Delta}{(n^\beta + c)^2}.
\end{aligned}
$$

Substituting for $\Delta$ in the r.h.s. of the above equation, we have

$$\mathbb{E}_\mathbb{P}\|\check{\mu}_\lambda - \mu_\mathbb{P}\|^2 - \Delta = \frac{(nc^2 + c^2 + 2cn^\beta)\|\mu_\mathbb{P}\|^2 - (c^2 + 2cn^\beta)\int k(x,x)\,\mathrm{d}\mathbb{P}(x)}{n(n^\beta + c)^2}.$$

It is easy to verify that $\mathbb{E}_\mathbb{P}\|\check{\mu}_\lambda - \mu_\mathbb{P}\|^2 - \Delta < 0$ if

$$\frac{\|\mu_\mathbb{P}\|^2}{\int k(x,x)\,\mathrm{d}\mathbb{P}(x)} < \inf_n \frac{c^2 + 2cn^\beta}{nc^2 + c^2 + 2cn^\beta} = \frac{2^{1/\beta}\beta}{2^{1/\beta}\beta + c^{1/\beta}(\beta-1)^{(\beta-1)/\beta}}.$$

*Remark.* If $k(x,y) = \langle x, y \rangle$, then it is easy to check that $\mathcal{P}_{c,\beta} = \{\mathbb{P} \in M_+^1(\mathbb{R}^d) : \frac{\|\theta\|_2^2}{\mathrm{trace}(\Sigma)} < \frac{A}{1-A}\}$ where $\theta$ and $\Sigma$ represent the mean vector and covariance matrix. Note that this choice of kernel yields a setting similar to classical James-Stein estimation, wherein for all $n$ and all $\mathbb{P} \in \mathcal{P}_{c,\beta} := \{\mathbb{P} \in \mathcal{N}_{\theta,\sigma} : \|\theta\| < \sigma\sqrt{dA/(1-A)}\}$, $\check{\mu}_\lambda$ is admissible for any $d$, where $\mathcal{N}_{\theta,\sigma} := \{\mathbb{P} \in M_+^1(\mathbb{R}^d) : \mathrm{d}\mathbb{P}(x) = (2\pi\sigma^2)^{-d/2}e^{-\frac{\|x-\theta\|^2}{2\sigma^2}}\,\mathrm{d}x,\ \theta \in \mathbb{R}^d,\ \sigma > 0\}$. On the other hand, the James-Stein estimator is admissible for only $d \geq 3$ but for any $\mathbb{P} \in \mathcal{N}_{\theta,\sigma}$.

## 2 Consequence of Theorem 1 if $k$ is translation invariant

**Claim:** Let $k(x,y) = \psi(x - y)$, $x, y \in \mathbb{R}^d$ where $\psi$ is a bounded continuous positive definite function with $\psi \in L^1(\mathbb{R}^d)$. For $\lambda = cn^{-\beta}$ with $c > 0$ and $\beta > 1$, define

$$\mathcal{P}_{c,\beta,\psi} := \left\{ \mathbb{P} \in M_+^1(\mathbb{R}^d) : \|\phi_{\mathbb{P}}\|_{L^2} < \sqrt{\frac{A(2\pi)^{d/2}\psi(0)}{\|\psi\|_{L^1}}} \right\},$$

where $\phi_{\mathbb{P}}$ is the characteristic function of $\mathbb{P}$. Then $\forall n$ and $\forall \mathbb{P} \in \mathcal{P}_{c,\beta,\psi}$, we have $\mathbb{E}_{\mathbb{P}}\|\check{\mu}_\lambda - \mu_{\mathbb{P}}\|^2 < \mathbb{E}_{\mathbb{P}}\|\hat{\mu}_{\mathbb{P}} - \mu_{\mathbb{P}}\|^2$.

*Proof.* If $k(x,y) = \psi(x - y)$, it is easy to verify that

$$\int\int k(x,y)\, d\mathbb{P}(x)\, d\mathbb{P}(y) = \int |\phi_{\mathbb{P}}(\omega)|^2 \widehat{\psi}(\omega)\, d\omega \le \sup_{\omega \in \mathbb{R}^d} \widehat{\psi}(\omega)\|\phi_{\mathbb{P}}\|_{L_2}^2 \le (2\pi)^{-d/2}\|\psi\|_{L_1}\|\phi_{\mathbb{P}}\|_{L_2}^2,$$

where $\widehat{\psi}$ is the Fourier transform of $\psi$. On the other hand, since $|\phi_{\mathbb{P}}(\omega)| \le 1$ for any $\omega \in \mathbb{R}^d$, we have

$$\int\int k(x,y)\, d\mathbb{P}(x)\, d\mathbb{P}(y) = \int |\phi_{\mathbb{P}}(\omega)|^2 \widehat{\psi}(\omega)\, d\omega \le \int |\phi_{\mathbb{P}}(\omega)|\widehat{\psi}(\omega)\, d\omega \le \|\phi_{\mathbb{P}}\|_{L^2}\|\widehat{\psi}\|_{L^2}$$

$$\le \|\phi_{\mathbb{P}}\|_{L^2}\sqrt{\|\widehat{\psi}\|_\infty\|\widehat{\psi}\|_{L^1}} = \|\phi_{\mathbb{P}}\|_{L^2}\sqrt{(2\pi)^{-d/2}\|\psi\|_{L^1}\psi(0)},$$

where we used $\psi(0) = \|\widehat{\psi}\|_{L^1}$. As $\int k(x,x)\, d\mathbb{P}(x) = \psi(0)$, we have that

$$\frac{\|\mu_{\mathbb{P}}\|^2}{\int k(x,x)\, d\mathbb{P}(x)} \le \min\left\{ \frac{\|\phi_{\mathbb{P}}\|_{L^2}^2\|\psi\|_{L^1}}{(2\pi)^{d/2}\psi(0)}, \sqrt{\frac{\|\phi_{\mathbb{P}}\|_{L^2}^2\|\psi\|_{L^1}}{(2\pi)^{d/2}\psi(0)}} \right\}.$$

Since $\mathbb{P} \in \mathcal{P}_{c,\beta,\psi}$, we have $\mathbb{P} \in \mathcal{P}_{c,\beta}$ and therefore the result follows. ∎

## 3 Proof of Theorem 2

Since $(e_i)_i$ is an orthonormal basis in $\mathcal{H}$, we have for any $\mathbb{P}$ and $f^* \in \mathcal{H}$

$$\mu_{\mathbb{P}} = \sum_{i=1}^\infty \mu_i e_i, \quad \hat{\mu}_{\mathbb{P}} = \sum_{i=1}^\infty \hat{\mu}_i e_i, \quad \text{and} \quad f^* = \sum_{i=1}^\infty f_i^* e_i,$$

where $\mu_i := \langle \mu_{\mathbb{P}}, e_i \rangle$, $\hat{\mu}_i := \langle \hat{\mu}_{\mathbb{P}}, e_i \rangle$, and $f_i^* := \langle f^*, e_i \rangle$. If follows from the Parseval's identity that

$$\Delta = \mathbb{E}_{\mathbb{P}}\|\hat{\mu} - \mu\|^2 = \mathbb{E}_{\mathbb{P}}\left[\sum_{i=1}^\infty (\hat{\mu}_i - \mu_i)^2\right] =: \sum_{i=1}^\infty \Delta_i$$

$$\Delta_{\boldsymbol{\alpha}} = \mathbb{E}_{\mathbb{P}}\|\hat{\mu}_{\boldsymbol{\alpha}} - \mu\|^2 = \mathbb{E}_{\mathbb{P}}\left[\sum_{i=1}^\infty (\alpha_i f_i^* + (1 - \alpha_i)\hat{\mu}_i - \mu_i)^2\right] =: \sum_{i=1}^\infty \Delta_{\boldsymbol{\alpha},i}.$$

Note that the problem has not changed and we are merely looking at it from a different perspective. To estimate $\mu_{\mathbb{P}}$, we may just as well estimate its Fourier coefficient sequence $\mu_i$ with $\hat{\mu}_i$. Based on above decomposition, we may write the risk difference $\Delta_{\boldsymbol{\alpha}} - \Delta$ as $\sum_{i=1}^\infty (\Delta_{\boldsymbol{\alpha},i} - \Delta_i)$. We can thus ask under which conditions on $\boldsymbol{\alpha} = (\alpha_i)$ for which $\Delta_{\boldsymbol{\alpha},i} - \Delta_i < 0$ uniformly over all $i$.

For each coordinate $i$, we have

$$\begin{aligned}
\Delta_{\boldsymbol{\alpha},i} - \Delta_i &= \mathbb{E}_{\mathbb{P}}\left[(\alpha_i f_i^* + (1 - \alpha_i)\hat{\mu}_i - \mu_i)^2\right] - \mathbb{E}_{\mathbb{P}}\left[(\hat{\mu}_i - \mu_i)^2\right] \\
&= \mathbb{E}_{\mathbb{P}}[\alpha_i^2 f_i^2 + 2\alpha_i f_i^*(1 - \alpha_i)\hat{\mu}_i + (1 - \alpha_i)^2\hat{\mu}_i^2 \\
&\quad -2\alpha_i f_i^*\mu_i - 2(1 - \alpha_i)\hat{\mu}_i\mu_i + \mu_i^2] - \mathbb{E}_{\mathbb{P}}[\hat{\mu}_i^2 - 2\hat{\mu}_i\mu_i + \mu_i^2] \\
&= \alpha_i^2 f_i^2 + 2\alpha_i f_i^*\mathbb{E}_{\mathbb{P}}[\hat{\mu}_i] - 2\alpha_i^2 f_i^*\mathbb{E}_{\mathbb{P}}[\hat{\mu}_i] + (1 - \alpha_i)^2\mathbb{E}_{\mathbb{P}}[\hat{\mu}_i^2] \\
&\quad -2\alpha_i f_i^*\mu_i - 2(1 - \alpha_i)\mathbb{E}_{\mathbb{P}}[\hat{\mu}_i]\mu_i + \mu_i^2 - \mathbb{E}_{\mathbb{P}}[\hat{\mu}_i^2] + 2\mu_i\mathbb{E}_{\mathbb{P}}[\hat{\mu}_i] - \mu_i^2 \\
&= \alpha_i^2 f_i^2 - 2\alpha_i^2 f_i^*\mu_i + (1 - \alpha_i)^2\mathbb{E}_{\mathbb{P}}[\hat{\mu}_i^2] - 2(1 - \alpha_i)\mu_i^2 + 2\mu_i^2 - \mathbb{E}_{\mathbb{P}}[\hat{\mu}_i^2] \\
&= \alpha_i^2 f_i^2 - 2\alpha_i^2 f_i^*\mu_i + (\alpha_i^2 - 2\alpha_i)\mathbb{E}_{\mathbb{P}}[\hat{\mu}_i^2] + 2\alpha_i\mu_i^2.
\end{aligned}$$

Next, we substitute $\mathbb{E}_{\mathbb{P}}[\hat{\mu}_i^2] = \mathbb{E}_{\mathbb{P}}[(\hat{\mu}_i - \mu_i + \mu_i)^2] = \Delta_i + \mu_i^2$ into the last equation to obtain

$$
\begin{aligned}
\Delta_{\boldsymbol{\alpha},i} - \Delta_i &= \alpha_i^2 f_i^2 - 2\alpha_i^2 f_i^* \mu_i + \alpha_i^2(\Delta_i + \mu_i^2) - 2\alpha_i(\Delta_i + \mu_i^2) + 2\alpha_i\mu_i^2 \\
&= \alpha_i^2 f_i^2 - 2\alpha_i^2 f_i^* \mu_i + \alpha_i^2 \Delta_i + \alpha_i^2 \mu_i^2 - 2\alpha_i \Delta_i \\
&= \alpha_i^2(f_i^2 - 2f_i^* \mu_i + \Delta_i + \mu_i^2) - 2\alpha_i \Delta_i \\
&= \alpha_i^2(\Delta_i + (f_i^* - \mu_i)^2) - 2\alpha_i \Delta_i
\end{aligned}
$$

which is negative if $\alpha_i$ satisfies

$$
0 < \alpha_i < \frac{2\Delta_i}{\Delta_i + (f_i^* - \mu_i)^2}.
$$

This completes the proof.

## 4  Proof of Proposition 3

Let $\mathbf{K} = \mathbf{U}\mathbf{D}\mathbf{U}^\top$ be an eigen-decomposition of $\mathbf{K}$ where $\mathbf{U} = [\tilde{\mathbf{u}}_1, \tilde{\mathbf{u}}_2, \ldots, \tilde{\mathbf{u}}_n]$ consists of orthogonal eigenvectors of $\mathbf{K}$ such that $\mathbf{U}^\top \mathbf{U} = \mathbf{I}$ and $\mathbf{D} = \mathrm{diag}(\tilde{\gamma}_1, \tilde{\gamma}_2 \ldots, \tilde{\gamma}_n)$ consists of corresponding eigenvalues. As a result, the coefficients $\boldsymbol{\beta}(\lambda)$ can be written as

$$
\boldsymbol{\beta}(\lambda) = g_\lambda(\mathbf{K})\mathbf{K}\mathbf{1}_n = \mathbf{U}g_\lambda(\mathbf{D})\mathbf{U}^\top \mathbf{K}\mathbf{1}_n = \sum_{i=1}^n \tilde{\mathbf{u}}_i g_\lambda(\tilde{\gamma}_i)\tilde{\mathbf{u}}_i^\top \mathbf{K}\mathbf{1}_n. \tag{1}
$$

Using $\mathbf{K}\mathbf{1}_n = [\langle \hat{\mu}, k(x_1, \cdot)\rangle, \ldots, \langle \hat{\mu}, k(x_n, \cdot)\rangle]^\top$, we can rewrite (1) as

$$
\begin{aligned}
\boldsymbol{\beta}(\lambda) &= \sum_{i=1}^n \tilde{\mathbf{u}}_i g_\lambda(\tilde{\gamma}_i) \sum_{j=1}^n \tilde{u}_{ij}\langle \hat{\mu}, k(x_j, \cdot)\rangle \\
&= \sum_{i=1}^n \sqrt{\tilde{\gamma}_i}\tilde{\mathbf{u}}_i g_\lambda(\tilde{\gamma}_i) \left\langle \hat{\mu}, \frac{1}{\sqrt{\tilde{\gamma}_i}} \sum_{j=1}^n \tilde{u}_{ij}k(x_j, \cdot) \right\rangle,
\end{aligned}
$$

where $\tilde{u}_{ij}$ is the $j$th component of $\tilde{\mathbf{u}}_i$. Next, we invoke the relation between the eigenvectors of the matrix $\mathbf{K}$ and the eigenfunctions of the empirical covariance operator $\widehat{\mathcal{C}}_k$ in $\mathcal{H}$. That is, it is known that the $i$th eigenfunction of $\widehat{\mathcal{C}}_k$ can be expressed as $\tilde{\mathbf{v}}_i = (1/\sqrt{\tilde{\gamma}_i})\sum_{j=1}^n \tilde{u}_{ij}k(x_j, \cdot)$ [2]. Consequently,

$$
\left\langle \hat{\mu}, \frac{1}{\sqrt{\tilde{\gamma}_i}} \sum_{j=1}^n \tilde{u}_{ij}k(x_j, \cdot) \right\rangle = \langle \hat{\mu}, \tilde{\mathbf{v}}_i\rangle
$$

and we can write the Spectral-KMSE as

$$
\begin{aligned}
\hat{\mu}_\lambda &= \sum_{j=1}^n \left[ \sum_{i=1}^n \tilde{u}_{ij}\sqrt{\tilde{\gamma}_i}g_\lambda(\tilde{\gamma}_i)\langle \hat{\mu}, \tilde{\mathbf{v}}_i\rangle \right] k(x_j, \cdot) \\
&= \sum_{i=1}^n \sqrt{\tilde{\gamma}_i}g_\lambda(\tilde{\gamma}_i)\langle \hat{\mu}, \tilde{\mathbf{v}}_i\rangle \sum_{j=1}^n \tilde{u}_{ij}k(x_j, \cdot) \\
&= \sum_{i=1}^n g_\lambda(\tilde{\gamma}_i)\tilde{\gamma}_i\langle \hat{\mu}, \tilde{\mathbf{v}}_i\rangle \tilde{\mathbf{v}}_i.
\end{aligned}
$$

This completes the proof.

## 5  Population counterpart of Spectral-KMSE

To obtain the population version of the Spectral-KMSE, we resort to the regression perspective of the kernel mean embedding which has been studied earlier in [3, 4]. The proof techniques used here are similar to those in [3]. Consider

$$
\arg\min_{\mathbf{F}\in\mathcal{H}\otimes\mathcal{H}} \quad \mathbb{E}_X \left[ \|k(X, \cdot) - \mathbf{F}k(X, \cdot)\|_{\mathcal{H}}^2 \right] + \lambda\|\mathbf{F}\|_{HS}^2. \tag{2}
$$

where $\mathbf{F} : \mathcal{H} \to \mathcal{H}$ is Hilbert-Schmidt. We can expand the regularized loss (2) as

$$\mathbb{E}_X \left[ \|k(X, \cdot) - \mathbf{F}k(X, \cdot)\|_{\mathcal{H}}^2 \right] + \lambda \|\mathbf{F}\|_{HS}^2$$
$$= \mathbb{E}_X \langle k(X, \cdot), k(X, \cdot) \rangle_{\mathcal{H}} - 2\mathbb{E}_X \langle k(X, \cdot), \mathbf{F}k(X, \cdot) \rangle_{\mathcal{H}} + \mathbb{E}_X \langle \mathbf{F}k(X, \cdot), \mathbf{F}k(X, \cdot) \rangle_{\mathcal{H}} + \lambda \langle \mathbf{F}, \mathbf{F} \rangle_{HS}$$
$$= \mathbb{E}_X \langle k(X, \cdot), k(X, \cdot) \rangle_{\mathcal{H}} - 2\mathbb{E}_X \langle k(X, \cdot) \otimes k(X, \cdot), \mathbf{F} \rangle_{HS} + \mathbb{E}_X \langle k(X, \cdot), \mathbf{F}^*\mathbf{F}k(X, \cdot) \rangle_{\mathcal{H}} + \lambda \langle \mathbf{F}, \mathbf{F} \rangle_{HS}$$
$$= \mathbb{E}_X \langle k(X, \cdot), k(X, \cdot) \rangle_{\mathcal{H}} - 2\langle \mathcal{C}_k, \mathbf{F} \rangle_{HS} + \langle \mathcal{C}_k, \mathbf{F}^*\mathbf{F} \rangle_{HS} + \lambda \langle \mathbf{F}, \mathbf{F} \rangle_{HS},$$

where $\mathbf{F}^*$ denotes the adjoint of $\mathbf{F}$ and $\mathcal{C}_k = \mathbb{E}_X[k(X, \cdot) \otimes k(X, \cdot)]$. Next, we show that the solution to the above expression is $\mathbf{F} := \mathcal{C}_k(\mathcal{C}_k + \lambda \mathbf{I})^{-1}$. Defining $\mathbf{A} := \mathbf{F}(\mathcal{C}_k + \lambda \mathbf{I})^{1/2}$, the above expression can be rewritten as

$$\mathbb{E}_X \langle k(X, \cdot), k(X, \cdot) \rangle_{\mathcal{H}} - 2\langle \mathcal{C}_k, \mathbf{F} \rangle_{HS} + \langle \mathcal{C}_k, \mathbf{F}^*\mathbf{F} \rangle_{HS} + \lambda \langle \mathbf{F}, \mathbf{F} \rangle_{HS}$$
$$= \mathbb{E}_X \langle k(X, \cdot), k(X, \cdot) \rangle_{\mathcal{H}} - 2\langle \mathcal{C}_k, \mathbf{F} \rangle_{HS} + \langle \mathcal{C}_k + \lambda \mathbf{I}, \mathbf{F}^*\mathbf{F} \rangle_{HS}$$
$$= \mathbb{E}_X \langle k(X, \cdot), k(X, \cdot) \rangle_{\mathcal{H}} - 2\langle \mathcal{C}_k, \mathbf{F} \rangle_{HS} + \left\langle \mathbf{F}(\mathcal{C}_k + \lambda \mathbf{I})^{1/2}, \mathbf{F}(\mathcal{C}_k + \lambda \mathbf{I})^{1/2} \right\rangle_{HS}$$
$$= \mathbb{E}_X \langle k(X, \cdot), k(X, \cdot) \rangle_{\mathcal{H}} - 2\langle \mathcal{C}_k, \mathbf{A}(\mathcal{C}_k + \lambda \mathbf{I})^{-1/2} \rangle_{HS} + \langle \mathbf{A}, \mathbf{A} \rangle_{HS}$$
$$= \mathbb{E}_X \langle k(X, \cdot), k(X, \cdot) \rangle_{\mathcal{H}} - \left\| \mathcal{C}_k(\mathcal{C}_k + \lambda \mathbf{I})^{-1/2} \right\|_{HS}^2 + \left\| \mathcal{C}_k(\mathcal{C}_k + \lambda \mathbf{I})^{-1/2} - A \right\|_{HS}^2.$$

As a result, the above expression is minimized when $\mathbf{A} = \mathcal{C}_k(\mathcal{C}_k + \lambda \mathbf{I})^{-1/2}$, implying that $\mathbf{F} = \mathcal{C}_k(\mathcal{C}_k + \lambda \mathbf{I})^{-1}$. As in the sample case, a natural estimate of the Spectral-KMSE is

$$\mu_\lambda = \mathbf{F}\mu_{\mathbb{P}} = \mathcal{C}_k(\mathcal{C}_k + \lambda \mathbf{I})^{-1}\mu_{\mathbb{P}}.$$

# 6  Proof of Proposition 4

The proof employs the relation between the Gram matrix $\mathbf{K}$ and the empirical covariance operator $\widehat{\mathcal{C}}_k$ shown in Lemma 3. It is known that the operator $\widehat{\mathcal{C}}_k$ is of finite rank, self-adjoint, and positive. Moreover, its spectrum has only finitely many nonzero elements [5]. If $\tilde{\gamma}_i$ is a nonzero eigenvalue and $\tilde{\mathbf{v}}_i$ is the corresponding eigenfunction of $\widehat{\mathcal{C}}_k$, then the following decomposition holds

$$\widehat{\mathcal{C}}_k f = \sum_{i=1}^n \tilde{\gamma}_i \langle f, \tilde{\mathbf{v}}_i \rangle_{\mathcal{H}} \tilde{\mathbf{v}}_i, \quad \forall f \in \mathcal{H}.$$

Note that it may be that $k < n$ where $k$ is the rank of $\widehat{\mathcal{C}}_k$. In that case, the above decomposition still holds. Setting $f = \hat{\mu}$ and applying the definition of the filter function $g_\lambda$ to the operator $\widehat{\mathcal{C}}_k$ yield

$$\hat{\mu}_\lambda = \widehat{\mathcal{C}}_k g_\lambda(\widehat{\mathcal{C}}_k)\hat{\mu} = \sum_{i=1}^n g_\lambda(\tilde{\gamma}_i)\tilde{\gamma}_i \langle \hat{\mu}, \tilde{\mathbf{v}}_i \rangle_{\mathcal{H}} \tilde{\mathbf{v}}_i,$$

which is exactly the decomposition given in Lemma 3. This completes the proof.

# 7  Proof of Theorem 5

Consider the following decomposition

$$\begin{aligned}
\hat{\mu}_\lambda - \mu_{\mathbb{P}} &= \widehat{\mathcal{C}}_k g_\lambda(\widehat{\mathcal{C}}_k)\hat{\mu}_{\mathbb{P}} - \mu_{\mathbb{P}} \\
&= \widehat{\mathcal{C}}_k g_\lambda(\widehat{\mathcal{C}}_k)(\hat{\mu}_{\mathbb{P}} - \mu_{\mathbb{P}}) + \widehat{\mathcal{C}}_k g_\lambda(\widehat{\mathcal{C}}_k)\mu_{\mathbb{P}} - \mu_{\mathbb{P}} \\
&= \widehat{\mathcal{C}}_k g_\lambda(\widehat{\mathcal{C}}_k)(\hat{\mu}_{\mathbb{P}} - \mu_{\mathbb{P}}) + (\widehat{\mathcal{C}}_k g_\lambda(\widehat{\mathcal{C}}_k) - I)\widehat{\mathcal{C}}_k^\beta h + (\widehat{\mathcal{C}}_k g_\lambda(\widehat{\mathcal{C}}_k) - I)(\mathcal{C}_k^\beta - \widehat{\mathcal{C}}_k^\beta)h
\end{aligned}$$

where we used the fact that there exists $h \in \mathcal{H}$ such that $\mu_{\mathbb{P}} = \mathcal{C}_k^\beta h$ as we assumed that $\mu_{\mathbb{P}} \in \mathcal{R}(\mathcal{C}_k^\beta)$ for some $\beta > 0$. Therefore

$$\|\hat{\mu}_\lambda - \mu_{\mathbb{P}}\| \le \|\widehat{\mathcal{C}}_k g_\lambda(\widehat{\mathcal{C}}_k)\|_{op}\|\hat{\mu}_{\mathbb{P}} - \mu_{\mathbb{P}}\| + \|(\widehat{\mathcal{C}}_k g_\lambda(\widehat{\mathcal{C}}_k) - I)\widehat{\mathcal{C}}_k^\beta\|_{op}\|h\| + \|\widehat{\mathcal{C}}_k g_\lambda(\widehat{\mathcal{C}}_k) - I\|_{op}\|\mathcal{C}_k^\beta - \widehat{\mathcal{C}}_k^\beta\|_{op}\|h\|$$

where we used the fact that $\|Ab\| \le \|A\|_{op}\|b\|$ with $A : \mathcal{H} \to \mathcal{H}$ being a bounded operator, $b \in \mathcal{H}$ and $\|\cdot\|_{op}$ denoting the operator norm defined as $\|A\|_{op} := \sup\{\|Ab\| : \|b\| = 1\}$.

By $(C1)$, $(C2)$ and $(C3)$, we have $\|\widehat{\mathcal{C}}_k g_\lambda(\widehat{\mathcal{C}}_k)\|_{op} \leq B$, $\|\widehat{\mathcal{C}}_k g_\lambda(\widehat{\mathcal{C}}_k) - I\|_{op} \leq C$ and $\|(\widehat{\mathcal{C}}_k g_\lambda(\widehat{\mathcal{C}}_k) - I)\widehat{\mathcal{C}}_k^\beta\|_{op} \leq D\lambda^{\min\{\beta,\eta_0\}}$ respectively. Denoting $\|h\| = \|\mathcal{C}_k^{-\beta}\mu_\mathbb{P}\|$, we therefore have

$$\|\hat{\mu}_\lambda - \mu_\mathbb{P}\| \leq B\|\hat{\mu}_\mathbb{P} - \mu_\mathbb{P}\| + D\lambda^{\min\{\beta,\eta_0\}}\|\mathcal{C}_k^{-\beta}\mu_\mathbb{P}\| + C\|\mathcal{C}_k^\beta - \widehat{\mathcal{C}}_k^\beta\|_{op}\|\mathcal{C}_k^{-\beta}\mu_\mathbb{P}\|. \qquad (3)$$

For $0 \leq \beta \leq 1$, it follows from Theorem 1 in [6] that there exists a constant $\tau_1$ such that

$$\|\mathcal{C}_k^\beta - \widehat{\mathcal{C}}_k^\beta\|_{op} \leq \tau_1\|\mathcal{C}_k - \widehat{\mathcal{C}}_k\|_{op}^\beta \leq \tau_1\|\mathcal{C}_k - \widehat{\mathcal{C}}_k\|_{HS}^\beta.$$

On the other hand, since $\alpha \mapsto \alpha^\beta$ is Lipschitz on $[0, \kappa^2]$ for $\beta \geq 1$, the following lemma yields that

$$\|\mathcal{C}_k^\beta - \widehat{\mathcal{C}}_k^\beta\|_{op} \leq \|\mathcal{C}_k^\beta - \widehat{\mathcal{C}}_k^\beta\|_{HS} \leq \tau_2\|\mathcal{C}_k - \widehat{\mathcal{C}}_k\|_{HS}$$

where $\tau_2$ is the Lipschitz constant of $\alpha \mapsto \alpha^\beta$ on $[0, \kappa^2]$. In other words,

$$\|\mathcal{C}_k^\beta - \widehat{\mathcal{C}}_k^\beta\|_{op} \leq \max\{\tau_1, \tau_2\}\|\mathcal{C}_k - \widehat{\mathcal{C}}_k\|_{HS}^{\min\{1,\beta\}}. \qquad (4)$$

**Lemma 1** (Contributed by Anreas Maurer, see Lemma 5 in [7]). *Suppose $A$ and $B$ are self-adjoint Hilbert-Schmidt operators on a separable Hilbert space $H$ with spectrum contained in the interval $[a, b]$, and let $(\sigma_i)_{i\in I}$ and $(\tau_j)_{j\in J}$ be the eigenvalues of $A$ and $B$, respectively. Given a function $r : [a, b] \to \mathbb{R}$, if there exists a finite constant $L$ such that*

$$|r(\sigma_i) - r(\tau_j)| \leq L|\sigma_i - \tau_j|, \ \forall i \in I, j \in J,$$

*then*

$$\|r(A) - r(B)\|_{HS} \leq L\|A - B\|_{HS}.$$

Using (4) in (3), we have

$$\|\hat{\mu}_\lambda - \mu_\mathbb{P}\| \leq B\|\hat{\mu}_\mathbb{P} - \mu_\mathbb{P}\| + D\lambda^{\min\{\beta,\eta_0\}}\|\mathcal{C}_k^{-\beta}\mu_\mathbb{P}\| + C\tau\|\mathcal{C}_k - \widehat{\mathcal{C}}_k\|_{HS}^{\min\{1,\beta\}}\|\mathcal{C}_k^{-\beta}\mu_\mathbb{P}\|, \qquad (5)$$

where $\tau := \max\{\tau_1, \tau_2\}$. We now obtain bounds on $\|\hat{\mu}_\mathbb{P} - \mu_\mathbb{P}\|$ and $\|\mathcal{C}_k - \widehat{\mathcal{C}}_k\|_{HS}$ using the following results.

**Lemma 2** ([8]). *Suppose that $\kappa = \sup_{x\in\mathcal{X}} \sqrt{k(x,x)}$. For any $\delta > 0$, the following inequality holds with probability at least $1 - e^{-\delta}$*

$$\|\hat{\mu}_\mathbb{P} - \mu_\mathbb{P}\| \leq \frac{2\kappa + \kappa\sqrt{2\delta}}{\sqrt{n}}.$$

**Lemma 3** (e.g., see Theorem 7 in [5]). *Let $\kappa := \sup_{x\in\mathcal{X}} \sqrt{k(x,x)}$. For $n \in \mathbb{N}$ and any $\delta > 0$, the following inequality holds with probability at least $1 - 2e^{-\delta}$:*

$$\left\|\widehat{\mathcal{C}}_k - \mathcal{C}_k\right\|_{HS} \leq \frac{2\sqrt{2}\kappa^2\sqrt{\delta}}{\sqrt{n}}.$$

Using Lemmas 2 and 3 in (5), for any $\delta > 0$, with probability $1 - 3e^{-\delta}$, we obtain

$$\|\hat{\mu}_\lambda - \mu_\mathbb{P}\| \leq \frac{2\kappa B + \kappa B\sqrt{2\delta}}{\sqrt{n}} + D\lambda^{\min\{\beta,\eta_0\}}\|\mathcal{C}_k^{-\beta}\mu_\mathbb{P}\| + C\tau\frac{(2\sqrt{2}\kappa^2\sqrt{\delta})^{\min\{1,\beta\}}}{n^{\min\{1/2,\beta/2\}}}\|\mathcal{C}_k^{-\beta}\mu_\mathbb{P}\|.$$

# 8 Shrinkage parameter $\lambda = cn^{-\beta}$

In this section, we provide supplementary results that demonstrate the effect of the shrinkage parameter $\lambda$ presented in Theorem 1. That is, if we choose $\lambda = cn^{-\beta}$ for some $c > 0$ and $\beta > 1$, the estimator $\check{\mu}_\lambda$ is a proper estimator of $\mu$. Unfortunately, the true value of $\beta$, which characterizes the smoothness of the true kernel mean $\mu_\mathbb{P}$, is not known in practice. Nevertheless, we provide simulated experiments that illustrate the convergence of the estimator $\check{\mu}_\lambda$ for different values of $c$ and $\beta$.

The data-generating distribution used in this experiment is identical to the one we consider in our previous experiments on synthetic data. That is, the data are generated as follows: $x \sim$

Figure 1: The risk of shrinkage estimator $\check{\mu}_\lambda$ when $\lambda = cn^{-\beta}$. The left figure shows the risk of the shrinkage estimator as sample size increases while fixing the value of $\beta$, whereas the right figure shows the same plots while fixing the value of $c$. See text for more explanation.

$\sum_{i=1}^{4} \pi_i \mathcal{N}(\boldsymbol{\theta}_i, \Sigma_i) + \varepsilon, \theta_{ij} \sim \mathcal{U}(-10, 10), \Sigma_i \sim \mathcal{W}(3 \times \mathbf{I}_d, 7), \varepsilon \sim \mathcal{N}(0, 0.2 \times \mathbf{I}_d)$ where $\mathcal{U}(a, b)$ and $\mathcal{W}(\Sigma_0, df)$ are the uniform distribution and Wishart distribution, respectively. We set $\boldsymbol{\pi} = [0.05, 0.3, 0.4, 0.25]$. We use the Gaussian RBF kernel $k(x, x') = \exp(-\|x - x'\|^2 / 2\sigma^2)$ whose bandwidth parameter is calculated using the median heuristic, i.e., $\sigma^2 = \text{median}\{\|x_i - x_j\|^2\}$. Figure 1 depicts the comparisons between the standard kernel mean estimator and the shrinkage estimators with varying values of $c$ and $\beta$.

As we can see in Figure 1, if $c$ is very small or $\beta$ is very large, the shrinkage estimator $\check{\mu}_\lambda$ behaves like the empirical estimator $\hat{\mu}_\mathbb{P}$. This coincides with the intuition given in Theorem 1. Note that the value of $\beta$ specifies the smoothness of the true kernel mean $\mu$ and is unknown in practice. Thus, one of the interesting future directions is to develop procedure that can adapt to this unknown parameter automatically.