[Reviews · NeurIPS 2014]

Submitted by Assigned_Reviewer_13

This paper deals with the estimation of the kernel mean, an effective tool to design kernel-based learning algorithms for probability measures. Specifically, the authors propose the use of spectral filtering to obtain estimators of the kernel mean better than the standard empirical estimator. Their estimator builds upon recent work [13] which demonstrates that the commonly used kernel mean estimator can be improved by shrinkage estimators. The authors provide a theoretical analysis of their spectral kernel mean estimator from an operator learning point of view and conducted artificial and real data experiments to test their spectral-based mean estimation under various settings and compare it with the standard empirical estimator and other shrinkage estimators.

The paper is well written and the subject is well motivated in the introduction. The paper clearly explains a new mechanism that should have margin to improve kernel mean estimation. The underlying idea is quite simple, but it is proved through a rigorous analysis and an empirical evaluation.

In the experimental section, it is true that Figure 2 shows that the spectral estimator is less expensive than other shrinkage estimator in terms of computational cost; however, it is much more expensive than the standard estimator. The data used in the experiments seem to be relatively small. The computational efficiency of the proposed estimation for large data is not clear.

Minor comment:
- line 87: better than \mu —> better than \hat{\mu}
- line 142: There is not (iii) in theorem 1

Pros:
- new kernel mean estimator
- well written paper
- the proposed method is justified by a theoretical analysis and experimental results

Cons:
- The data used in the experiments seem to be relatively small
Summary: This is a nice and an interesting approach for estimating the kernel mean.

Submitted by Assigned_Reviewer_28

This paper addresses the stein effect when using empirical average to estimate the kernel mean that had been pointed by by [13]. [13] first proposed a simple shrinkage estimator to solve this issue. In this paper, the authors propose a systematic way to obtain shrinkage estimator for kernel mean, using the spectral filtering, that has been widely used in kernel based learning. It is not surprised to see several well-known iterated algorithms for spectral filtering being introduced for kernel mean estimation.

Another contribution of this paper is the theoretical analysis for this shrinkage estimator provided by this paper. It connects the F-KMSE in [13] to their spectral filtering framework and provides a concentration bound for this estimator. And the result roughly matches the usual optimal rate, (n^{1/2}), for this kind of kernel estimators.

Moreover, the authors provide experiments on both synthetic and real data to demonstrate their points in the paper. The evaluation on the real data seems a little bit complicated, since the loss function is the negative log-likelihood of a mixed distribution Q fit from the estimated kernel density using their algorithms. I understand that they use this two-layer method is because the distribution for real data is unknown and the evaluation is hard. But it inevitably includes the extra error caused by the mixture distribution estimation. The experiments on real data could be stronger if they come up with more intuitive measure.
Summary: This paper introduces a systematic way to estimate of the kernel mean, using spectral filtering, and also provides theoretical analysis for the spectral filtered estimator of kernel mean.

Submitted by Assigned_Reviewer_42

The paper presents a family of kernel mean shrinkage estimators. These estimators generalize the ones proposed in [13] and can incoporate useful domain knowledge through spetral filters. Here is a summary of interesting contributions:
1. Theorem 1 that shows the consistency and admissibility of kmse presented in [13].
2. The idea of spectral kmse (its use in this unsupervised setting) and similarity of final form with the supervised setting.
3. Theorem 5 that shows consistency of the proposed spectral kmse.

Comments:
1. Overall the paper is well written and presents a few interesting results as mentioned above.
2. However the empirical results are not at all convincing:
a. Fig 3 gives rather mixed results. Only early stopping effect seems clear.
b. Table 2: With such close values, I doubt if any of these are statistically significant improvements. Also, it will be nice to see some cases where the improvement is considerable.
3. Owing to the above perhaps the wording in the abstract and conclusions perhaps should avoid superlatives like outperform etc.

Minor Comments:
1. F-KMSE is not visible in fig.3c.
2. Though it can be inferred from [13] and context, it will be nice to clarify what S-KMSE, F-KMSE etc. represent.
Summary: The paper has interesting results but a very weak empirical section, which when improved warrants a publication.
Author Feedback
Author rebuttal: We thank all reviewers for their reviews. Below we respond to the major concerns. Our revision however will also address the remaining minor comments.

We first address the meta-review: "Could you please highlight the potential impact of your work considering previous estimators of kernel mean?"

Kernel mean estimation is central to kernel methods in that it is used by classical approaches (e.g., when centering a kernel PCA matrix), and it also forms the core inference step of modern kernel methods that rely on embedding probability distributions in RKHSes (e.g., kernel independence testing or kernel two-sample testing relies on kernel mean embedding).

Up until [13], researchers always used the standard mean estimator. The paper [13] showed that shrinkage can help, but in a way that did not sufficiently take into account the specifics of kernel methods. The present paper studies this, and connects it to spectral filtering.

Specifically:

(1) We propose a more general family of shrinkage estimators which broadens the applications of shrinkage estimators of the kernel mean. Previous work [13] focused only on the empirical aspects of S-KMSE and F-KMSE. In this work, we prove the consistency and admissibility of the S-KMSE and show that our estimators include F-KMSE as a special case. Most theoretical analyses such as our consistency result in Theorem 5 also apply to the F-KMSE of [13].

(2) From a practical point of view, our empirical studies suggest that the proposed estimators will be very useful for "large p, small n" situations, which are common in a number of real-world applications (e.g. medical data, gene expression analysis, and text documents). Additionally, the choice of filter function may allow the incorporation of available prior information.

(3) This work broadens the application of spectral filtering algorithms in the unsupervised setting. We believe this may eventually lead to a better understanding of the fundamental relationship between Tikhonov regularization and Stein’s shrinkage estimation in RKHSes. That is, while the classical spectral filtering approach extends Tikhonov regularization from the supervised learning perspective (e.g., classification and regression), this work gives a seemingly related extension of Stein’s shrinkage estimation (e.g., location and scale parameters estimation) in an unsupervised setting.

Below we respond to each reviewer individually.

== Reviewer 13 ==

You are correct that all shrinkage estimators are more computationally expensive than the standard estimator, especially when sample size is large. On the other hand, with increase in sample size, the shrinkage estimator behaves similarly to that of the standard estimator and therefore the incentive to use a shrinkage estimator decreases. Also, as mentioned above, the most interesting cases are those in which few high-dimensional observations are available, i.e., the "large p, small n" setting. Hence, the computational cost becomes less important, as the algorithm for learning the weights only scales with sample size n rather than dimension p.

== Reviewer 28 ==

The results reported in Table 2 should be robust to extra error caused by the mixture model estimation because, in each experiment, we initialized the mixture model by the best model obtained from the K-means algorithm with 50 initializations. The negative log-likelihood scores are then averaged over 30 repetitions of the experiment.

== Reviewer 42 ==

- Re. the results in Fig 3: the figure is not trying to show an advantage over early stopping. We consider Landweber or Iterated Tikhonov as examples of suitable spectral filtering, see also Table 1. The main point of Fig 3 is that (a) shrinkage in F-KMSE or S-KMSE can achieve the same result as early stopping, (b) iterative approaches can be faster than F-KMSE, and (c) as the dimensionality p increases, shrinkage helps, and the methods that take into account the spectrum do best. We understand we need to be more explicit about this in the caption.

- The paper is not doing the usual thing of proposing a method and then showing that it works better than the rest. Rather, we provide an analysis of a family of methods, and Fig 3 illustrates some aspects of how they behave (including, in particular, that they do better than the standard mean estimator). We agree we should weaken the wording of abstract and conclusions to reflect this.

- Small improvements in Table 2 may actually be considerable in term of likelihood itself because the scores are on a log-scale, i.e., negative log-likelihood, and we often observed a very small variance. To address your concern, we can conduct statistical tests such as a pairwise t-test to check whether the improvements are statistically significant and incorporate the results into our revision.